# Monocyte to HDL and Neutrophil to HDL Ratios as Potential Ischemic Stroke Prognostic Biomarkers

**Aimilios Gkantzios** [1], **Dimitrios Tsiptsios** [1,*], **Vaia Karapepera** [1], **Stella Karatzetzou** [1], **Stratis Kiamelidis** [1],
**Pinelopi Vlotinou** [1], **Erasmia Giannakou** [1], **Evangeli Karampina** [2], **Katerina Paschalidou** [2],
**Nikolaos Kourkoutsakis** [3], **Nikolaos Papanas** [4], **Nikolaos Aggelousis** [2] and **Konstantinos Vadikolias** [1]

[1]  Neurology Department, Democritus University of Thrace, 68100 Alexandroupolis, Greece
[2]  Department of Physical Education and Sport Science, Democritus University of Thrace,
     69100 Komotini, Greece
[3]  Radiology Department, Democritus University of Thrace, 68100 Alexandroupolis, Greece
[4]  Second Department of Internal Medicine, Democritus University of Thrace, 68100 Alexandroupolis, Greece
*   Correspondence: tsiptsios.dimitrios@yahoo.gr; Tel.: +30-6944320016

**Abstract:** Ischemic stroke (IS) exhibits significant heterogeneity in terms of etiology and pathophysiology. Several recent studies highlight the significance of inflammation in the onset and progression of IS. White blood cell subtypes, such as neutrophils and monocytes, participate in the inflammatory response in various ways. On the other hand, high-density lipoproteins (HDL) exhibit substantial anti-inflammatory and antioxidant actions. Consequently, novel inflammatory blood biomarkers have emerged, such as neutrophil to HDL ratio (NHR) and monocyte to HDL ratio (MHR). Literature research of two databases (MEDLINE and Scopus) was conducted to identify all relevant studies published between 1 January 2012 and 30 November 2022 dealing with NHR and MHR as biomarkers for IS prognosis. Only full-text articles published in the English language were included. Thirteen articles have been traced and are included in the present review. Our findings highlight the utility of NHR and MHR as novel stroke prognostic biomarkers, the widespread application, and the calculation of which, along with their inexpensive cost, make their clinical application extremely promising.

**Keywords:** monocyte to HDL-ratio; neutrophil to HDL-ratio; MHR; NHR; stroke prognosis; stroke outcome; blood biomarkers

## 1. Introduction

Despite substantial advancements in preventative measures and therapeutic therapies, stroke continues to be the second greatest cause of mortality in people and accounts for the majority of acquired disability cases [1–3]. The socioeconomic burden of stroke survivors will increase at an unprecedented rate due to the age-related character of the disease, the fact that more than half of patients are over 65 [3–5], the growing worldwide population and the ongoing improvement in life expectancy. Therefore, it is crucial to identify patients with poor prognoses early and precisely in order to customize treatment and rehabilitation to meet the needs of each individual [6–8].

Ischemic stroke (IS) exhibits significant heterogeneity both in terms of its etiology and pathophysiology, as well as its clinical manifestation. Although tremendous progress has been made in the fields of acute ischemic stroke (AIS) therapy and imaging, numerous steps are still required in the areas of prognosis and post-stroke rehabilitation [9,10]. Along with the use of various clinical assessment tools, as well as neurophysiological techniques, the identification and development of new biomarkers are essential, particularly during the acute period of IS. In this direction, the use of blood biomarkers appears to be very promising, given their immediate availability in almost every health facility and their low cost [4,11–13].

Certain characteristics should be present in an effective biomarker. According to the Biomarkers Definition Working Group, a biomarker is a physiological characteristic or biological substance that can be evaluated objectively and can serve as an indicator of either normal or pathological biological processes, as a risk factor, or as a pharmacological response to therapeutic intervention, with high sensitivity and specificity [4,14–16]. Furthermore, to be clinically applicable, every biomarker should be readily available, easy to reproduce, and noninvasively obtained from the available sources, cost-effective, indicative of the pathophysiological process and easily interpretable by the clinician [4,17].

Inflammation plays a significant role in the development, evolution, and prognosis of IS. There is growing evidence indicating that inflammation contributes to the immune response in the affected site, promoting brain cells' death and exacerbating neurological dysfunction [18,19]. Inflammation is involved not only in the pathophysiology of atherosclerosis but also in the activation of the vascular endothelium; it contributes to the rupture of the blood–brain barrier (BBB) and mediates the oxidative procedure, as well as the infiltration of leukocytes and platelets, which results in post-ischemia injuries [20–22].

A frequent indicator of the inflammatory process is the total number of white blood cells (WBC) [23]. A high WBC count at admission has been positively associated with higher severity, worse prognosis and increased IS mortality in several case-control studies [24–27]. Different WBC subtypes, including neutrophils, lymphocytes, and monocytes, take part in the inflammatory response. Neutrophils have a significant role in various stages of atherosclerosis and contribute to the instability of atherosclerotic plaque by escalating endothelial dysfunction, luring monocytes, activating macrophages, and promoting the production of foam cells [28–31]. By producing proinflammatory cytokines that make the atherosclerotic plaque unstable and trigger sequelae such as rupture of the atherosclerotic plaque, bleeding, or thrombosis, the main group of inflammatory cells, monocytes, participate in all stages of the inflammatory response [24,32–34]. On the contrary, the reformation and cessation of the inflammatory process are related to the function of lymphocytes. A growing body of research indicates a connection between a low lymphocyte count, a greater infarct volume and a worse neurological functional result [21,35]. Additionally, a high monocyte count is associated with the development of atherosclerotic plaque. Monocytes are one of the most crucial elements of the inflammatory process in atherosclerosis. An important phase in the development of atherosclerotic plaques is the migration of monocytes into the intima, where they differentiate into macrophages, which through the use of their scavenger receptors absorb oxidized low-density lipoprotein (ox-LDL) [36,37]. Recruited monocytes go through activation and concentrate as foam cells as a reaction to fatty plaques. Inside this multi-step adhesion cascade, endothelial cell-expressed selectins facilitate the initial monocyte-endothelium attachment. Firm attachment happens when CD11b/CD18 and other monocyte adhesion molecules connect with the vascular cell adhesion molecule (VCAM)-1 and intracellular adhesion molecule-1 (ICAM-1). VCAM-1, intracellular adhesion molecule (ICAM)-1, E-selectin, and pro-leukocyte extravasation cytokines such as IL-8 are all inhibited by HDL. HDL is believed to play a significant anti-inflammatory and antioxidant role by controlling the activation of monocytes, preventing macrophage migration and inhibiting the oxidation of low-density lipoprotein (LDL) by blocking the 12-lipoxygenase that produces lipid hydroperoxides and leads to the oxidation of the LDL via the transition of metal ions. Given the above, HDL protects the endothelium of blood arteries from the detrimental effects of LDL [38,39].

Novel inflammatory blood biomarkers have arisen as a result of the aforementioned characteristics, and they may be utilized in routine clinical practice. Neutrophils, monocytes, lymphocytes and HDL are common hematological markers that are affordable, easy to assess, and widely used. The equilibrium between proatherogenic and antiatherogenic variables is represented by the monocyte-to-high-density-lipoprotein cholesterol ratio (MHR) and a high ratio is associated with the severity of atherosclerosis. Both the neutrophil-to-HDL ratio (NHR) and the monocyte-to-HDL ratio (MHR) are potential in-

flammatory markers that suggest endothelial dysfunction, atherosclerosis and thrombosis, making them desirable clinical predictors of the prognosis of IS [40–42].

## 2. Materials and Methods

The Preferred Reporting Items for Systematic Reviews and Meta-analyses (PROSPERO registration number: CRD42023390050) was used to guide this study. Our study's methods were a priori designed.

### 2.1. Literature Search

Literature research of two databases (MEDLINE and Scopus) was conducted by two investigators (AG and VK) in order to trace all relevant studies published between 1 January 2012, and 30 November 2022 using the terms ["white blood cells to HDL ratio" OR "neutrophil to HDL ratio" OR "lymphocyte to HDL ratio" OR "monocyte to HDL ratio" OR "high-density lipoprotein" OR "HDL" OR "monocyte" OR "lymphocyte" OR "neutrophil" OR "white blood cells"] AND ["stroke prognosis" OR "stroke outcome" or "stroke recovery"] as keywords. The retrieved articles were also manually searched for any further potential eligible articles. Any disagreement regarding the screening or selection process was solved by a third investigator (KV) until a consensus was reached.

### 2.2. Eligibility Criteria

Only full-text original articles published in English were included. Secondary analyses, reviews, guidelines, meeting summaries, comments, unpublished abstracts, or studies conducted on animals were excluded. There were no restrictions on study design or sample characteristics.

### 2.3. Data Extraction

Data extraction was performed using a predefined data form created in Excel. We recorded the author, year of publication, biomarker, type of stroke, type of study, number of participants and their mean age, time of blood sampling, scale(s) of stroke severity and prognosis/clinical outcome, cut-off values and finally, the main results of the study.

### 2.4. Data Analysis

No statistical analysis or meta-analysis was performed due to the high heterogeneity among the studies. Thus, the data were only descriptively analyzed.

## 3. Results

### 3.1. Database Searches

Overall, 625 records were retrieved from the database search. Duplicates and irrelevant studies were excluded. Then, 181 records were screened with secondary search criteria and 41 full-text articles were selected as potentially eligible for our review. After screening the full text of the articles, 13 studies were ultimately eligible for inclusion (Figure 1).

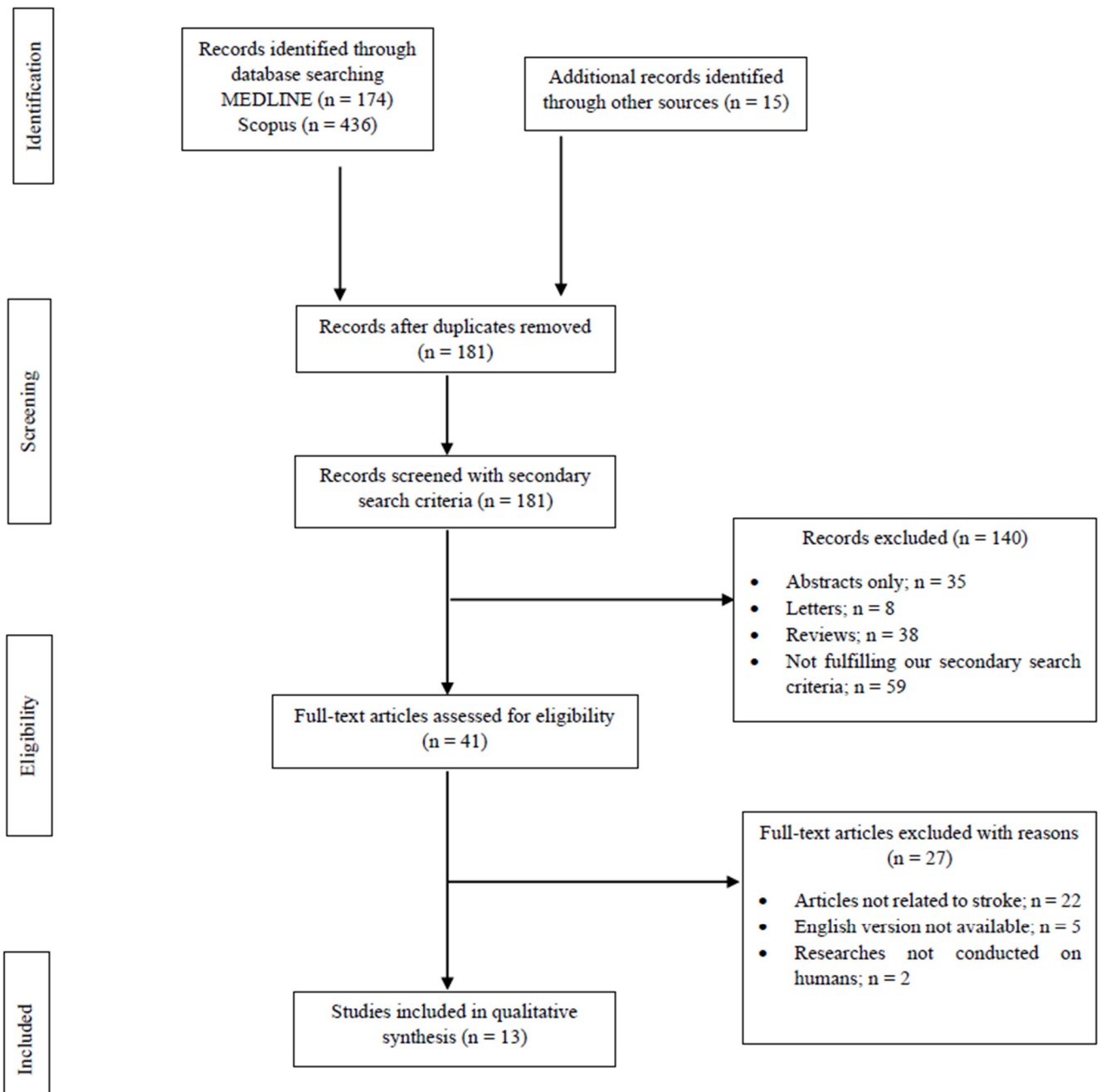

**Figure 1.** Study flowchart (PRISMA diagram).

### 3.2. Study Characteristics

Thirteen publications fulfilled our inclusion criteria. They were classified based on the primary biomarker they examined. Hence, two of the studies focused on the NHR and the rest eleven studies deal with the prognostic value of MHR (Table 1).

### 3.3. Study Design

All thirteen were longitudinal, either retrospective or prospective.

**Table 1.** Basic characteristics of the 13 included studies.

| Authors, Year of Publication | Biomarker | Type of Study | Type of Stroke | Number of Participants/ Mean Age | Time of Blood Sampling | Scale of Stroke Severity and Prognosis/Clinical Outcome | Cut-Off Values; (Specificity); [Sensitivity] | Main Results |
|---|---|---|---|---|---|---|---|---|
| 1. Chen et al., 2020 [43] | NHR | Retrospective | IS | 160 patients, 160 healthy controls/patients: 67.67 ± 12.08, healthy controls: 67.62 ± 10.38 | Within 24 h of admission | NIHSS, mRS | NHR > 5.66; (79.6%); [51.6%] | The NHR was strongly connected with neurological impairment and 3-month outcomes in AIS patients |
| 2. Jiang et al., 2022 [44] | NHR | Prospective | IS (as part of cardiovascular events in general) | 34,335/49.6 ± 18.2 | Not mentioned | None | Not mentioned | In the general population, NHR was independently correlated with cardiovascular and all-cause death |
| 3. Bolayir et al., 2017 [45] | MHR | Retrospective | IS | 466 IS patients, 408 controls/IS patients: 77.09 ± 6.7, controls: 77.35 ± 9.6 | Within 24 h of admission | None | MHR > 17.52; (84.8%); [94.4%] | In IS patients, a high MHR value at admission may be a reliable indicator of 30-day mortality |
| 4. Algin et al., 2019 [25] | MHR | Retrospective | IS | 75 patients/73.23 ± 11.49 | On admission | GCS, NIHSS | 0.191; (90%); [52.3%] | Short-term mortality was highly correlated with MHR |
| 5. Liu et al., 2020 [46] | MHR | Retrospective | IS | 253 patients, 211 healthy Subjects/Patients:67, healthy controls:66 years | Within 24 h of admission | NIHSS | 0.28 with an area under the curve: 0.777; (77.25%); [66.01%] | MHR is characterized as a distinct risk factor for the development of IS. Together, MHR and MLR showed increased sensitivity for the IS diagnosis |
| 6. Liu et al., 2020 [47] | MHR | Prospective | IS | 1090 patients/ Patients with good outcome: 59.32 ± 12.64, patients with poor outcome: 64.13 ± 12.21 | Within 24 h of admission | mRS | 0.51; (66.5%); [62.3%] | In AIS patients, MHR may be a substantial and independent indicator of poor functional prognosis |
| 7. Wang et al., 2020 [48] | MHR | Retrospective | IS | 974 patients/69 (IQR: 58–78) | Within 24 h of admission | NIHSS | Not mentioned | In individuals with IS, reduced MHR was independently linked to an increased risk of HT and symptomatic HT |
| 8. Oh et al., 2020 [49] | MHR and LMR | Retrospective | IS (patients with LAO treated with MT) | 411/68.6 | On admission, before MT | mRS, NIHSS | MHR cutoff: 1.4; N.A. | Higher MHR and NLR, and lower LMR values were detected after MT in patients who had a poor outcome. Scores based on inflammation, such as the MHR, NLR and LMR might be independent factors of patients' clinical outcome and prognosis after MT |

**Table 1.** *Cont.*

| Authors, Year of Publication | Biomarker | Type of Study | Type of Stroke | Number of Participants/ Mean Age | Time of Blood Sampling | Scale of Stroke Severity and Prognosis/Clinical Outcome | Cut-Off Values; (Specificity); [Sensitivity] | Main Results |
|---|---|---|---|---|---|---|---|---|
| 9. Bi et al., 2021 [50] | MHR | Retrospective | IS | 212 patients/68.27 ± 11.57 | Within 24 h of admission | NIHSS | MHR cutoff 0.51; (76.6%); [65.5%] | As a biomarker of END in individuals with isolated pontine infarction, elevated MHR may be useful, and the higher MHR was independently linked to the END |
| 10. Sun et al., 2021 [51] | MHR | Retrospective | IS | 803 patients/69 ± 12.3 | Within 24 h of admission | NIHSS, mRS | Not mentioned | In patients with AIS, elevated MHR and MC on admission are both linked to SAP, but not to all-cause mortality at 3 months |
| 11. Li et al., 2021 [52] | MHR | Retrospective | IS | 316 patients/64.66 ± 12.24 | Within 24 h of admission | mRS | Not mentioned | They proposed that poor 3-month functional outcomes in LAA IS were dependently associated with greater MHR values |
| 12. Li et al., 2021 [53] | MHR | Retrospective | IS | 286 patients/70.00 (IQR: 63.00–77.00) | All blood tests were performed before MT | NIHSS, mRS | 0.368; (56.8%); [67.4%] | In patients with AIS who received MT, higher RPR, MHR, and NLR may be independent risk factors for predicting a poor outcome at three months |
| 13. Xia et al., 2022 [54] | MHR | Retrospective | IS | 340 patients/69.5 ± 13.5 | At 24 h after thrombolytic treatment | NIHSS | 0.46; (57.1%); [70.6%] | In patients with acute ischemic stroke receiving intravenous thrombolysis, elevated MHR may be independently linked to a greater risk of HT |

Abbreviations: AIS: Acute ischemic stroke, END: early neurological deterioration, GCS: Glasgow Coma Scale, HT: Hemorrhagic transformation, IQR: interquartile ranges, IS: ischemic stroke, IS: Ischemic stroke, IVT: intravenous thrombolysis, LAA: large artery atherosclerosis, LAO: large artery occlusion, LMR: lymphocyte to monocyte ratio, MC: Monocytes, MHR: Monocytes to HDL ratio, mRS: modified Rankin Scale, MT: Mechanical thrombectomy, N.A.: not applicable, NIHSS: National Institutes of Health Stroke Scale, NHR: Neutrophil to HDL ratio, NLR: neutrophil-to-lymphocyte ratio, RPR: RDW to platelet ratio, SAP: stroke-associated pneumonia.

### 3.4. Stroke Patients Group

The total number of IS patients included in all studies ranges from *n* = 75 [25] to *n* = 1090 [47]. Across the 13 studies, one study has a disease sample size between 1–100 patients, one study between 101–200, three studies between 201–300, two studies between 301–400, five studies have a disease sample size larger than 400 patients, whereas one study conducted their examination in a pool of 34,335 individuals amongst the general population.

### 3.5. Reference Groups

Across the 13 studies, stroke patients are contrasted to demographically matched healthy individuals in only two studies, with the rest of them not including a healthy control group. One of the studies included a disease-control group other than stroke patients, notably they were age and sex-matched and gave blood samples for reasons other than AIS. These patients did not suffer from any uncontrolled systemic diseases such as diabetes mellitus, hypertension, cardiovascular diseases, or cancer. However, one study focused on examining a large pool of individuals from the general population and not only stroke patients.

### 3.6. Demographic and Clinical Profiles

Mean/median patients' age ranges from $49.6 \pm 18.2$ years [44] to $77.09 \pm 6.7$ years [45]. 10 studies examined patients with IS, one study patients with acute isolated pontine infarction, one study Large Artery Atherosclerosis IS and one study examined the incidence of all kinds of death and cardiovascular mortality (including IS) in the general population.

### 3.7. Time of Blood Sampling

In two studies blood sampling was performed upon admission, in nine studies in the first 24 h, in one study before thrombectomy and in one study the time of blood sampling was not mentioned.

### 3.8. Scales of Stroke Severity and Prognosis/Clinical Outcome

National Institutes of Health Stroke Scale (NIHSS) was used in four studies, modified Rankin Scale (mRS) in two, both NIHSS and mRS in four, and both NIHSS and Glasgow Coma Scale (GCS) in one, whereas two studies did not utilize any scale.

## 4. Discussion

A literature review over the last decade was conducted focusing on the potential prognostic value of NHR and MHR after IS. Thirteen studies were included; in two the NHR was the subject of their study, and in eleven it was the MHR. NHR is a composite marker of inflammation and lipid metabolism, whereas MHR has emerged as a novel inflammation marker and has been highly linked to cardiovascular events.

### 4.1. Neutrophils-to-HDL-Ratio (NHR)

Regarding NHR, Chen et al. [43] retrospectively sought the connection between NHR and AIS in patients undergoing intravenous thrombolysis (IVT) and evaluated its correlation with the severity and prognosis of IS. First, they examined demographic and laboratory features (such as age, sex, neutrophil, HDL-C and NHR) of IS patients compared with those of healthy controls; then, they divided stroke patients into three groups based on their 3-month prognosis in association with NHR: Q1, NHR < 3.65; Q2, $3.65 \leq$ NHR $\leq 5.59$; and Q3, NHR > 5.59. Their research revealed that the Q3 group, namely those with the higher NHR levels, had greater rates of NIHSS admission, 24-h NIHSS, 7-day NIHSS and 3-month mRS, as well as higher neutrophil counts and incidence of hyperlipidemia. In addition, up to 25% of patients with NHR > 5.59 may exhibit severe neurological impairments. Furthermore, a number of variables, including age, atrial fibrillation and smoking are found to be correlated with poor outcomes at the same time. Finally, using the ROC curve to determine

the accuracy of predicting the 3-month prognosis of IS patients, the NHR cutoff value was determined to be 5.66, with a sensitivity of 51.6% and a specificity of 79.0%. Therefore, the authors concluded that due to the interaction between increased neutrophils and decreased HDL-C in AIS patients, NHR may enhance the risk of AIS via inflammatory activity and aberrant lipid metabolism.

In a broader context, using clinical data from the National Health and Nutrition Examination Survey, Jiang et al. [44] conducted the first study assessing the predictive validity of NHR for all-cause and cardiovascular mortality, including IS, as well as its predictive utility for the long-term clinical outcomes among the general U.S. adult population (NHANES). Their sample was selected to be representative of the general community and the subsequent study comprised 34,335 participants. Through questionnaires, they gathered demographic information about the participants, including age, sex, race/ethnicity, level of education, smoking and alcohol use, as well as information about their medical history, including diabetes mellitus, hypertension, heart failure, coronary heart disease, stroke, cancer and body mass index (BMI). After calculating the NHR, statistical analysis using a fully adjusted Cox regression model revealed that participants in the highest tertile had a 29% greater risk of death from any cause than those in the lowest tertile, whereas the middle tertile did not significantly differ from the lowest tertile. In addition, there were large differences in the frequency of cardiovascular fatalities between the tertiles of NHR. Following further subgroup analysis, participants in the highest tertile had a greater risk of all-cause death than those in the lowest tertile, apart from those aged <60 and those with diabetes. In addition, there were statistically significant interactions between NHR and sex for all-cause mortality, indicating that the link between NHR and all-cause mortality was higher for female participants than for male participants. There were no significant associations between NHR and cardiovascular mortality; however, there was a significant relationship between NHR and cardiovascular mortality among older women, those with hypertension, and those without diabetes. The researchers concluded that the relationship between NHR and all causes of mortality was U-shaped and not linear, whereas the relationship between NHR and cardiovascular disease death was linear.

### 4.2. Monocyte-to-HDL Ratio (MHR)

Concerning MHR, Bolayir et al. [45] examined it as a blood biomarker predictive of stroke progression. They assessed 466 patients with IS who presented to their clinic within 24 h of the onset of symptoms and compared them to 408 controls. Patients with IS were divided into two groups based on 30-day death rates. The first group consisted of survivors, while the second consisted of individuals who died within 30 days of a stroke. Patients' blood samples were taken within the first 24 h following AIS. In addition, the patients in the control group were matched for age and sex and provided samples for reasons other than AIS. These individuals had no chronic systemic disorders, including diabetes, hypertension, cardiovascular disease, or cancer. A comparison of baseline demographic parameters between the patient and control groups indicated no significant differences in terms of age, sex, diabetes mellitus status, hypertension status, BMI values, statin use and smoking status. In a comparison of laboratory measures between the patient and control groups, there was no substantial difference in the levels of creatinine, hemoglobin, LDL, or total cholesterol. On the other hand, uric acid, glucose, mean platelet volume (MPV), C-reactive protein (CRP), platelet, monocyte, lymphocyte, neutrophil and white blood cell (WBC) counts, HDL and MHR values differed significantly between the two groups. While the disease group had higher uric acid, glucose, MPV, CRP, platelet, monocyte, neutrophil and WBC counts and MHR values, the control group had considerably higher lymphocyte and HDL levels. In the two subgroups of the patients, comparing the monocyte count, HDL level and MHR value of the two groups revealed that the second group (those who died within 30 days after AIS) had significantly higher monocyte count and MHR value, while the first group (those who survived) had statistically lower HDL levels ($p < 0.001$). The independent variables age, sex, presence of diabetes and hypertension, cigarette smoking, HDL, LDL,

total cholesterol, hemoglobin, CRP, MHR and uric acid were utilized to determine risk factors associated with 30-day mortality in AIS patients. Multivariate analysis revealed that uric acid, CRP and MHR levels were statistically significant independent variables for predicting 30-day death in AIS patients. In addition, the optimal MHR cutoff value for predicting 30-day death in AIS patients was determined using receiver operating characteristic (ROC) analysis. With a sensitivity of 94.4% and a specificity of 84.5%, the cutoff MHR value was determined to be 17.52.

In their study, Algin et al. [25] evaluated 75 IS patients retrospectively. In order to predict a patient's prognosis one month after a stroke, the researchers divided their sample of patients into two groups based on whether or not death happened within one month. They utilized NIHSS score, stroke volume, neutrophil, platelet, WBC, albumin, monocyte/HDL ratio, Ca+ and Glasgow coma scale as predictors of mortality. Following ROC analysis, each of these characteristics predicted one-month mortality with *p* values of 0.001, 0.003, 0.01, 0.046, 0.007, 0.024, 0.047, 0.035, and 0.003, respectively. None of the following characteristics revealed a statistically significant relationship with mortality after multivariate analysis, indicating that additional research is required. Regarding MHR, Algin et al. demonstrated that MHR is related to short-term mortality, and they calculated a cut-off value of 0.19 with a sensitivity of 52.3% and a specificity of 90%.

Additionally, Liu et al. [46] retrospectively evaluated 253 patients with IS and 211 healthy controls to determine the diagnostic utility of MHR, MLR (monocyte to lymphocyte ratio) and their combination in IS patients. In this study, the link between the aforementioned IS parameters was established for the first time. They collected baseline demographic data and vascular risk factors, including age, sex, hypertension, diabetes, hyperlipidemia, coronary heart disease (CHD), atrial fibrillation and medication history. Age, sex, hypertension, diabetes, hyperlipidemia, heart illness and medications did not differ significantly between the two groups, as determined by a comparison of their baseline demographic information. In addition, there were no significant variations in blood glucose, cholesterol (CHOL), or low-density lipoprotein cholesterol (LDL-C) levels between the two groups. However, the levels of triglycerides (TGs), monocytes and lymphocytes were significantly higher in the disease group compared to the control group, although the level of HDL-C was significantly lower. In addition, MHR and MLR levels were elevated in the disease group relative to the control group. Based on univariate logistic analysis MHR and MLR were identified as potential risk variables for ischemic stroke. MHR or MLR was an independent predictor of the existence of ischemic stroke with an adjusted OR of 1.043 and 1.093, respectively, and the patient group had higher MHR and MLR levels than the control group. In addition, they identified a mildly positive relationship between the MHR and NIHSS. Based on the examination of the ROC curve, the appropriate MHR and MLR cutoff values for diagnosing ischemic stroke were 0.2816 and 0.1958, respectively. The AUC of MHR was calculated to be 0.777, with a sensitivity of 66.0% and a specificity of 77.2%. Additionally, the AUC of MLR was calculated to be 0.742, with a sensitivity of 70.36 % and a specificity of 67.77 %. When MHR and MLR were combined to identify IS, sensitivity was 73.91 % and specificity was 74.4 %. This was a greater performance in predicting IS compared to MHR and MLR alone, particularly in terms of sensitivity, and these results may indicate an ongoing inflammatory process in the pathophysiology of IS.

Furthermore, Liu et al. [47] investigated the relationship between MHR and 3-month function prognosis in patients with IS in an effort to determine the MHR level that most reliably identifies patients who are at a higher risk for an unsatisfactory outcome and require further potential treatment. Based on their mRS, they divided the patients into two groups: those with a favorable outcome had an mRS < 2 and those with a negative prognosis had an mRS > 2. They documented the patient's demographic information, including age, sex, smoking and statin use history, as well as the patient's stroke risk factors, including stroke history, diabetes, hypertension, dyslipidemia, coronary heart disease and atrial fibrillation. A multivariate logistic regression analysis was performed to determine the relationship between MHR and prognosis. The receiver operating characteristic (ROC)

analysis was then performed to analyze the predictive ability of the MHR, monocytes and HDL. After adjusting for the aforementioned demographic features and risk factors, the multivariable logistic analysis revealed that the MHR was independently related to the 3-month poor outcome. Monocytes remained substantially related, but not HDL. Compared to the good outcome group, patients with poor outcomes were older, female, had a higher NIHSS score at admission and had a history of smoking, statin use and stroke. Finally, the ROC analysis and area under the curve (AUC) revealed that the MHR had the highest AUC value compared to monocytes and HDL, and the best predictive cutoff value was 0.51 with a sensitivity of 62.3% and a specificity of 66.6%.

An increasing number of studies suggest that the post-stroke inflammatory response and its effects on the BBB exacerbate hemorrhagic transformation (HT) [55]. Meanwhile, on the one hand, the primary class of inflammatory cells, monocytes, invade the ischemic region and release proinflammatory cytokines that exacerbate brain damage [56,57]. On the other hand, the BBB can be protected by HDL by preventing monocytes from adhering to endothelial cells and by decreasing the oxidation of LDL. In accordance with the above, Wang et al. [48] decided to examine the relationship between MHR and HT in IS patients for the first time. They recorded and compared the characteristics of patients with and without HT. Those with HT were much older and had a significantly reduced number of males. On admission, the HT group had a significantly higher NIHSS score as well as a higher prevalence of atrial fibrillation and a larger infarct size. HT patients also exhibited decreased systolic blood pressure, monocyte count, triglyceride and low-density lipoprotein cholesterol levels, but higher HDL. Those in the HT group were less likely to receive antiplatelet and lipid-lowering medications, but more likely to undergo thrombolysis and thrombectomy. They found that patients with HT were shown to have a considerably lower MHR than those without HT. Similar outcomes were reported regardless of whether participants had symptomatic HT. In multivariate logistic regression analysis, the MHR was inversely linked with HT as a continuous variable. When MHR was divided into tertiles (Tertile 1: < 0.22; Tertile 2: 0.22 to 0.37; and Tertile 3: > 0.37), the crude OR of HT in the lowest tertile was 1.85 in comparison to the highest tertile. Even after correcting for variables, the lowest tertile group maintained a significantly higher risk of HT than the highest tertile group. The lowest tertile of MHR was associated with a 3.82-fold increase in symptomatic risk after adjustment for NIHSS and substantial infarct volume. Further research utilizing restricted cubic spline regression revealed a connection between a higher MHR and a lower chance of HT and symptomatic HT. The association between MHR and HT was unaffected by age (<60 vs. ≥60), sex (male vs. female), atrial fibrillation, stroke subtype (cardioembolic vs. non-cardioembolic), alcohol consumption, baseline NIHSS score (<15 vs. ≥15), baseline systolic blood pressure (<140 vs. ≥140) and low-density lipoprotein cholesterol (<2.52 vs. ≥2.52). Wang et al. concluded, based on their study, that a lower MHR was independently linked with increased risks of HT, particularly symptomatic HT following AIS and that the MHR may serve as a promising marker for identifying patients with a higher risk of HT and supporting physicians in selecting an appropriate treatment to prevent bleeding hazards.

Towards the same direction, Xia et al. [54], questioned if patients with AIS receiving IVT would benefit from using the MHR as a predicting factor for HT. Consequently, they conducted a study to investigate whether IVT is associated with MHR and HT and in what manner. Fifteen percent of the 340 participants included in their study experienced HT. HT was linked with significantly higher rates of atrial fibrillation, admission NIHSS score, white blood cell count, monocyte count and international normalized ratio (INR). Furthermore, bridging treatment was substantially more prevalent among HT patients. The aforementioned were also confirmed by the Crude models. The median MHR for all patients was 0.44 (0.31–0.59) and HT was linked with a significantly higher MHR. After correcting for relevant confounders, multivariate logistic regression indicated MHR as an independent risk factor for HT. Whether MHR was analyzed as a continuous variable or a quartile variable, the same result was achieved. With a sensitivity of 70.6% and specificity

of 57.1%, MHR was able to distinguish between patients in their sample who experienced HT and those who did not at the ideal cutoff of 0.46.

Moreover, the conventional treatment for acute ischemic stroke (AIS) caused by large artery occlusion (LAO) is mechanical thrombectomy (MT). Oh et al. [49] decided to evaluate the connection between the inflammatory process and the prognosis of patients who underwent MT because of LAO using inflammation-based scoring. MHR, NLR (neutrophil-to-lymphocyte ratio) and LMR (lymphocyte-to-monocyte ratio) were the ratios based on inflammation parameters that were utilized. On the grounds of laboratory data, inflammation-based scores, such as the NLR, the LMR and the MHR, were calculated for 411 patients retrospectively. Mortality, symptomatic intracranial hemorrhage (sICH), hemorrhagic transformation (HT) of infarct and unfavorable prognosis (mRS score of 3–6) were evaluated. Multivariate analyses were done to evaluate the correlations between inflammation-based scores and other clinical outcomes. Following the European Cooperative Acute Stroke Study [ECASS] III trial, all patients underwent IVT (intravenous thrombolysis) with t-PA (tissue plasminogen activator, alteplase) within 4.5 h of stroke onset at a maximum dose of 0.9 mg/kg. As determined by computed tomography angiography, LAO includes occlusion of the intracranial carotid artery (ICA), middle cerebral artery (MCA, M1, or M2), anterior cerebral artery (ACA) and posterior circulation (vertebral artery or basilar artery) (CTA). At 3 months, all patients were dichotomized based on mRS (favorable 0–2 vs. unfavorable 3–6). Receiver operating characteristic (ROC) analysis was carried out to determine connections between inflammation-based scores (NLR, LMR and MHR) and clinical outcomes and to identify suitable cutoff values for outcome prediction. Given a univariate study, correlations between variables and clinical outcomes were determined. Input into a backward multivariable logistic regression analysis were univariable analysis factors with $p < 0.20$. Multivariable regression was utilized to assess independent determinants of clinical outcomes after MT and to determine the odds ratio (OR) and 95% confidence interval (CI) for each endpoint. An unfavorable 3-month outcome (mRS score 3–6) was found for the other 199 patients. In the unfavorable group, the WBC and neutrophil counts were dramatically elevated. In contrast, the number of lymphocytes increased in the favorable group. The unfavorable group showed greater NLR and MHR levels but lower LMR values than the favorable group. Successful recanalization was more prevalent in patients with positive outcomes. In the unfavorable group, the prevalence of sICH, HT of infarct, and mortality were considerably higher than in the favorable group. ROC analysis identified an NLR of 5.1 as the best cutoff value for discriminating between favorable (mRS: 0–2) and unfavorable (mRS: 3–6) outcomes at 3 months. Using the same methodology, the optimal LMR cutoff for favorable mRS and MHR cutoff for unfavorable mRS were determined to be 2.5 and 1.4, respectively. In addition, both univariable and multivariable logistic regression analyses were carried out. A greater value of NLR ($\geq$5.1), a higher value of MHR ($\geq$1.4), and a lower value of LMR (<2.5) on admission were independently linked with a poor outcome. Other recognized characteristics, including being elderly, a high initial NIHSS, failure to achieve recanalization, and sICH, were also independently related to a poor prognosis. Higher NLR ($\geq$5.1) and the presence of sICH were separately related to higher mortality following MT. Higher NLR ($\geq$5.1) and lower LMR (<2.5) were independent predictors associated with elevated sICH. Furthermore, higher NLR ($\geq$5.1) and higher MHR ($\geq$1.4) were independently related to HT of the infarct. In summary, Oh et al. found that the mean MHR was higher among patients who underwent MT with a poor outcome than in those with a positive outcome, with a threshold value of 1.4. Moreover, an MHR value greater than 1.4 was an independent predictor of an unsatisfactory three-month prognosis and HT of the infarct.

Similarly, Li et al. [52], based on the knowledge that the production of proinflammatory chemokines that trigger a significant inflammatory process by brain tissue during IS is well recognized, chose to investigate the usefulness of MHR, NLR and RPR (RDW to platelet ratio) for the prognosis of IS in patients who underwent mechanical thrombectomy (MT). They retrospectively evaluated 286 IS patients who were treated with MT. Regarding

their prognosis, they separated those patients into two groups: those with a favorable prognosis at 3 months, as measured by mRS (0–2) and those with an unfavorable prognosis at 3 months as also measured by mRS (3–6). They reported that the unfavorable group had higher MHR, NLR and RPR levels than the favorable group and that the overall prevalence of 3-month mortality was considerably greater in the unfavorable group than the favorable. They also proposed the optimal cutoff values of RPR, MHR and NLR that predicted the 90-days prognosis of AIS patients undergoing MT. There were 8.565 (sensitivity: 51.4%, specificity: 93.2%), 0.368 (sensitivity: 67.4%, specificity: 56.8%) and 4.030 (sensitivity: 71.7%, specificity: 66.2%) respectively. Finally, they demonstrated that the prediction value of the three variables when combined is stronger than the prediction value of each one separately.

Up to one-third of people with AIS experience neurological impairments, often known as early neurological deterioration (END), which has been linked to higher mortality and functional disability [58–60]. Bi et al. [50], investigated the potential link between MHR and IS, specifically with isolated pontine infarction. They evaluated retrospectively 212 patients who had undergone an acute isolated pontine infarction to determine whether there was a linkage between MHR and END. END was defined as an increase in overall NIHSS 2 or an increase in motor power NIHSS 1 within the first week following stroke. Age, sex, and vascular risk factors such as diabetes mellitus, hypertension, coronary heart disease, hyperlipidemia, past stroke and smoking status were gathered. In order to test the probability of threshold effects, they examined the overall MHR by quartiles with increasing values. Patients were divided into the following quartiles based on the MHR: (Q1, <0.24, Q2, 0.24–0.42, Q3, 0.43–0.55, and Q4, ≥0.56). They determined that the average MHR was 0.44 ± 0.22 and that 27.36% of the 212 patients with isolated pontine infarction were diagnosed with END. In the majority of hospitalized patients (65.52%), END occurred within the first 48 h. According to Bi et al.'s multivariate analysis of the sample, which adjusted for confounding and risk factors, NIHSS at admission, basilar artery stenosis and fasting blood glucose were independently correlated with END. The odds ratio for END increased as the quartile level of MHR rose, with the lowest quartile serving as the standard. In multivariate analysis, the third and fourth quartiles of MHR were associated with 4.847-fold and 5.824-fold greater chances of END compared to the first quartile of MHR. They determined that END and raised MHR levels are associated and that the risk of END increased as MHR levels rose.

Since stroke-associated pneumonia (SAP) is prominent following a stroke and MHR is a novel inflammation marker closely connected with cardiovascular events which appear to be a promising biomarker for cerebrovascular events as well, Sun et al. [51] chose to investigate the interaction between MHR and SAP and 3-month mortality. They identified 803 AIS patients prospectively and gathered baseline data, including patient demographic characteristics, vascular risk factors, stroke severity (NIHSS and mRS), medication use, imaging data and diagnosis-related information. History of stroke, hypertension, diabetes mellitus, atrial fibrillation, coronary heart disease, present or past smoking status and alcohol intake were the risk factors. For further analysis of the sample, patients were classified into quartiles based on MHR (Q1: < 0.21, Q2: 0.21–0.30, Q3: 0.30–0.45, Q4 ≥ 0.45) and monocyte count (MC) (Q1: <0.30, 109/L; Q2: 0.30–0.40, 109/L; Q3: 0.40–0.50, 109/L; Q4: ≥0.50, 109/L) as measured upon admission. The correlations between quartiles of MHR and MC and the probability of SAP and all-cause mortality at 3 months were assessed using crude and multivariable logistic regression models that accounted for potential confounding variables. As for the predictive validity of MHR and MC on the risk of SAP, according to the unadjusted logistic regression model, SAP was substantially greater among research participants with admission MHR in the highest quartile (≥0.45), compared to those in the lowest quartile (≤0.20). After controlling for age, sex, baseline NIHSS score and other conventional risk variables, the OR (95% CI) for the highest quartile of MHR at admission was 2.79 (95% CI: 1.44–5.42; P trend = 0.003) in comparison to the lowest quartile of SAP. SAP was also substantially greater in the unadjusted logistic regression model among research participants with admission MC in the highest quartile (≥0.50) compared to those in the

lowest quartile (<0.30). After controlling for possible confounders, the association between increasing MC and SAP remained significant (OR 2.60, 95% CI 1.28–5.20, P trend = 0.005). Regarding the relationship between MHR, MC and 3-month mortality, the unadjusted logistic regression model analysis revealed no significant correlation between greater MHR and 3-month mortality. Moreover, after correcting for age, sex, baseline NIHSS score and other traditional risk factors, they did not discover a significant connection between MHR and 3-month mortality. In the uncorrected logistic regression model, study participants with admission MC in the highest quartile had a greater death rate than those in the lowest quartile. After controlling for potential confounders, the link between increased MC and 3-month mortality was not significant. As a result, Sun et al. concluded in their study that among patients with IS, a greater admission MHR was associated with SAP. However, there was no correlation between MHR and 3-month death from all causes.

According to the Trial of ORG 10172 Acute Stroke Treatment (TOAST) classification, ischemic stroke caused by large artery atherosclerosis (LAA) is a common type of ischemic stroke [61,62]. Li et al. [53] conducted a retrospective study on 316 LAA ischemic stroke patients in order to investigate the relationship between the MHR and the patient's 3-month functional prognosis. The scientists determined that the group with a poor functional outcome had a higher MHR level than the group with a good functional outcome. In addition, compared to patients with a favorable 3-month outcome, patients with a poor 3-month outcome were older, had a higher prevalence of diabetes mellitus, a higher baseline NIHSS score and a higher incidence of nosocomial infection. Patients with a poor prognosis showed greater white blood cell (WBC), neutrophil ($p = 0.015$), and fasting blood glucose (FBG) levels, but lower triglyceride levels. In addition, the MHR was separated into quartiles (lower tertile: MHR $\geq$ 0.03, but <0.32; middle tertile: MHR $\geq$ 0.32, but <0.48; upper tertile: MHR $\geq$ 0.48, but $\leq$1.21). To evaluate the relationship between MHR and 3-month functional prognosis, univariate and multivariate logistic regression analyses were conducted. They determined that the MHR was a substantial risk factor for the poor 3-month outcome of LAA IS. This connection similarly followed a positive linear pattern.

In the present review, one question was to which extent both NHR and MHR are influenced by sex. Unfortunately, this is only acknowledged indirectly in three of the thirteen studies, with no analysis of why this is the case. To the best of our knowledge, there is no direct pathophysiological reference in the bibliography that correlates either of the two ratios examined in the review at hand to sex. In general, women have higher concentrations of HDL cholesterol than men because estrogen appears to increase this good cholesterol. However, everything changes during menopause. At this point, the cholesterol levels of many women change: total and LDL cholesterol rise, while HDL cholesterol falls. Studies on stroke and poststroke prognosis indicate that in women, poststroke lipid levels appeared unrelated to outcome, whereas in males, greater triglycerides (TG) and LDL levels were associated with an improved prognosis. Another opinion supports the assumption that the risk of stroke related with lipid profiles is subtle and varies by stroke subtype, but does not consistently differ by sex [63,64]. As for neutrophils, females had 0.66 times more neutrophils than males on average in their peripheral blood. In addition, women under the age of 50 have a higher proportion of neutrophils and a lower proportion of lymphocytes than men, but this pattern is reversed in women over the age of 51. Neutrophils from male people contain more lipid mediators of inflammation, such as leukotrienes and prostaglandin E2. In addition, male neutrophils were less developed and had a higher activation threshold than female neutrophils of the same age, resulting in neutrophil-specific immune-metabolic signatures. Neutrophils increase in the brain after an ischemic stroke and correlate positively with stroke severity and infarct volume, as well as poststroke outcomes. Neutrophil activation and migration-specific genes were both time- and sex-dependent; immediately following cardioembolic stroke, females exhibited differentially expressed genes, whereas males did not. In contrast, in experimental models of stroke, older males had increased brain neutrophil infiltration than their female counterparts of the same age [65,66]. Unfortunately, our literature search could not reveal any relationship between

monocytes and sex. However, there is considerable evidence for a possible association between NHR and MHR and the sex of stroke patients based on the information presented above. Nonetheless, additional research is necessary to demonstrate their relationship with certainty.

*4.3. Limitations*

We recognize that our study has several limitations that must be taken into account. Firstly, most of the included studies have been single-centered, the majority of them with a small sample size. The aforementioned may hamper the generalizability of the study's findings and, therefore, lead to bias. Due to the intricacy of death classification in a clinical setting, the evaluation of all-cause mortality may influence the causal relationship between stroke and death events. Additionally, the current study did not investigate the impact of treatment, which could have an influence on the concentrations of biomarkers, or the effect of the duration of biomarker level testing. Finally, due to the variability of the results and the absence of cut-off values in four of the thirteen studies included in this review, we were unable to define a cut-off value for the ratios we decided to examine. To improve output consistency, it is required to decrease the aforementioned constraints.

## 5. Conclusions

This review examines the potential clinical applications of NHR and MHR as novel inflammatory biomarkers in stroke prognosis, taking into consideration all relevant factors. The widespread application and calculation of these parameters, along with their inexpensive cost, make the clinical application of these biomarkers extremely promising. To further elucidate this clinically significant relationship, it is recommended that further research should be conducted on the association between NHR and MHR levels and the odds of recovery in stroke patients. A comprehensive panel offers more stability and specificity than a single indicator; hence, additional multicenter trials with a bigger patient sample are required.

**Author Contributions:** A.G. and V.K. reviewed the literature, screened the abstracts of the reference list, deleted duplicates and citations not meeting the inclusion criteria, and assessed the articles; K.V. resolved any disagreements regarding screening or the selection process; A.G. wrote the first manuscript; S.K. (Stella Karatzetzou), S.K. (Stratis Kiamelidis), P.V., E.G., E.K., K.P., N.A. reviewed the tables, the presentation of the data, and the methodology. The corrected version was discussed collegially. N.K., N.P., A.G. and D.T. wrote the final version. All authors have read and agreed to the published version of the manuscript.

**Funding:** This work was supported by the project "Study of the interrelationships between neuroimaging, neurophysiological and biomechanical biomarkers in stroke rehabilitation (NEURO-BIO-MECH in stroke rehab)" (MIS 5047286), which is implemented under the action of "Support for Regional Excellence", funded by the operational program "Competitiveness, Entrepreneurship and Innovation" (NSRFm2014-2020) and co-financed by Greece and the European Union (the European Regional Development Fund).

**Institutional Review Board Statement:** Not applicable.

**Informed Consent Statement:** Not applicable.

**Data Availability Statement:** All data discussed within this manuscript are available on PubMed.

**Conflicts of Interest:** The authors declare no conflict of interest.

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
