# Peer review of "Monocyte to HDL and Neutrophil to HDL Ratios as Potential Ischemic Stroke Prognostic Biomarkers"

_2035-8377, doi:10.3390/neurolint15010019_

Round 1

Reviewer 1 Report

My opinion: 1. the research is interesting due to the involvement of inflammation in the formation and progression of ischemic stroke 2. Too few patients in particular groups 3. NHR and MHR may be new prognostic biomarkers.

Author Response

Dear Reviewer,

Many thanks for your kind words and the time spent reviewing our manuscript.

Your comment was added to the limitations of our study

"Firstly, most of the included studies have been single - centered with a small sample size the majority of them. The aforementioned may hamper the generalizability of the study's findings and, therefore, lead to bias. "

Looking forward to your follow-up comments.

Yours Sincerely,

Dr D Tsiptsios

Reviewer 2 Report

This is a very complete review of the literature however it is unfortunate that the authors do not try to better explain the discrepancies between the different studies, whose would be important to suggest an optimal MHR or NHR. Both the MHR and NHR recommendations are different between the previously published studies, the authors just described them but do not try to understand what make them different. It is therefore difficult to get an idea what would be an appropriate ratio. 

I also suggest to add a section describing how MHR and NHR might be influence by sex.

Finally the first point of limitation of the study is not necessary. Here the authors reviewed the literature, the sample size is therefore dependent of the previous published studies. The authors also do not suggest what would be the optimal MHR and NHR, therefore what exactly should be validated in larger scale studies? 

Author Response

Dear Reviewer,

Many thanks for your prompt response and the time spent reviewing our manuscript.

According to your suggestion a paragraph describing how MHR and NHR might be influenced by sex was added.

Moreover, the fact that we cannot suggest an optimal MHR or NHR was added to the limitations of our study.

Looking forward to your follow up comments.

Yours Sincerely,

Dr D Tsiptsios

Reviewer 3 Report

The article is a review paper dealing with novel inflammatory blood biomarkers in stroke prognosis: neutrophil to HDL ratio (NHR) and monocyte to HDL ratio (MHR). The authors searched two databases (MEDLINE and Scopus) for studies published between 1 January 2012 and 30 November 2022. In the end, they selected 13 articles. The authors concluded that these parameters, NHR and MHR, along with their inexpensive cost, make the clinical application of these biomarkers extremely promising.

The paper is well written and may be high interesting to the readers because of the novelity of the subject.

In my opinion it can be published in Neurology International. 

Author Response

Dear Reviewer,

Many thanks for your kind words and the time spent reviewing our manuscript.

Yours Sincerely,

Dr D Tsiptsios